# From ultrasound to microscopy: Actualities in thyroid investigation in cattle

**Justine Eppe** [1]*, **Elise Raguet**[1], **Patrick Petrossians**[2], **Sébastien Czaplicki**[3], **Calixte Bayrou**[1], **Frédéric Rollin**[1], **Vinciane Toppets**[4], **Hugues Guyot**[1]

1 Clinical Department of Production Animals, Fundamental and Applied Research for Animals & Health Research Unit (FARAH), Faculty of Veterinary Medicine, University of Liège, Liege, Belgium, **2** Department of Endocrinology, University Hospital of Liege, University of Liege, Liege, Belgium, **3** Department of Veterinary Management of Animal Resources, Fundamental and Applied Research for Animals & Health Research Unit (FARAH), University of Liege, Liege, Belgium, **4** Department of Morphology and Pathology, Faculty of Veterinary Medicine, University of Liege, Liege, Belgium

* justine.eppe@uliege.be

**Data Availability Statement:** All relevant data are contained in the manuscript and its supplementary information files.

**Funding:** The author(s) received no specific funding for this work.

## Abstract

Thyroid ultrasonography examination is widely used in human and small animal medicine. However, it has rarely been applied in cattle. The aim of this study was to determine whether the measurements of the thyroid gland by ultrasound examination correlate to those taken during *post-mortem* examination. A sample of 22 cows and 23 calves was selected for thyroid gland evaluation. An ultrasound scan was performed *ante-mortem*, followed by euthanasia (for medical reasons) or slaughtered in the food chain and the dissection of the thyroid gland was therefore performed. *Post-mortem*, the gland was weighed and its dimensions and volume measured. The volume and weight measurements were compared with the predicted ones on US using the formulas available in the literature. Finally, histological examination was performed on thyroid glands. The dimensions of the thyroid gland measured by ultrasonography were significantly different (p<0.05) from those observed *post-mortem*, except for lobe lengths in calves (p>0.1). However, in calves, there was no systematic bias between the ultrasound and *post-mortem* examination of the thyroid gland, which were concordant (with an average error of 18%). Cystic lesions were observed on ultrasound in 9/22 cows and could be found on histological examination in 7 of these. Other lesions, such as follicular hypoplasia and hyperplasia, were seen on histological examination but not on ultrasound. Although the ultrasound measurements did not significantly correlate with those taken *post-mortem*, this examination may allow to differentiate non-standard thyroids in the case of hyperplastic goiter, as demonstrated in other species. This study also describes and illustrates interesting lesions of the thyroid gland in cattle. These findings are innovative in the description of the use of thyroid ultrasound in cattle, although further studies are needed to allow deeper conclusions.

**Competing interests:** The authors have declared that no competing interests exist.

## Introduction

The importance of thyroid metabolism in cattle is widely recognized but tools to assess thyroid function in these animals are insufficient [1–4]. The thyroid gland can be examined using ultrasonography (US), an examination which can be carried out on a non-sedated animal and takes just a few minutes [1, 5]. Recently, we described an US procedure of the thyroid gland that was technically validated. The measurements obtained in this study had low intra-observer variability (calves: 10.4%; cows: 11.8%) [5].

US is the favorite ancillary examination for assessing the structure of the thyroid gland [6]. It is systematically used in human [6–8] and pet medicine [9–11], when thyroid disease is suspected. There are many conditions that can be detected by US examination, ranging from diffuse diseases such as chronic autoimmune thyroiditis (Hashimoto's thyroiditis) or Grave's diseases, to lesions in the thyroid parenchyma such as cysts or nodules [6–8, 12].

In ruminants, the literature describes lesions visible on thyroid glands removed *post-mortem* and reports the presence of cystic structures in buffaloes [13], cattle [14] and sheep [15]. Histological examination in cattle may reveal colloid goiter, parenchymal goiter, a combination of colloid and parenchymal goiter, fibrosis, lymphocytic thyroiditis, congestion and hemorrhage [13, 16]. To illustrate histological findings, a study describes the presence of hyperplastic goiter in cows that have undergone the Fukushima accident [17]. A hyperplastic goiter is associated with cysts in one study on two cows [14]. Cancers have also been described, such as thyroid carcinoma in goat [18] and cattle [19–21]; or thyroid fibrosarcoma in a sheep [22].

Further, diffuse thyroid diseases such as hypothyroidism [23–25] and hyperthyroidism [24, 25] are also described in cattle. The implication of thyroid metabolism in other diseases such as weak calf syndrome [26] or acute respiratory distress syndrome [27] is also documented.

The originality of our study is the connection between the macroscopic and microscopic evaluation of the thyroid lesions. Our first objective was to verify whether the measurements taken during the US investigation are concordant to *post-mortem* findings of the thyroid gland. Our second aim was to compare thyroid structural abnormalities observed during US with gross *post-mortem* lesions and their appearance on histopathological examination.

## Materials and methods

This research project has been accepted by the Animal Ethics Commission of the University of Liege under protocol n°2224.

For statistical purposes, a minimum of 20 calves and 20 cows that had to be euthanized for medical reasons (examined at the University Veterinary Clinic in Liege, Clinical Department of Production Animals) or healthy fattening cows in the slaughterhouse, were enrolled in this study. All breeds of cattle were admitted. The only condition was that the calves should be aged between 0 and 3 months old.

All animals had an *ante-mortem* clinical general examination [28] and a thyroid US (5). Three views were successively observed (left sagittal, right sagittal, transverse) to estimate the length, thickness, width of the two lobes and thickness of the isthmus of the thyroid gland. The US scanner used was a Mindray DP-50VET (Shenzhen, China) with a linear probe of 10 mHz with harmonics for the calves and a convex probe of 5 mHz for cows.

Thyroid volume was estimated using the formulae previously validated in humans (a) [7], dogs (a) [9] and goats (b) [29].

a. *length * thickness * width * π/4* (in cm$^3$; for each lobe)

b. *length * thickness * width * π/6* (in cm$^3$; for each lobe)

The expected *post-mortem* thyroid weight was obtained using the formula of Hernandez (1972): $Y = 0.348 * X^{0.944}$, where Y is the thyroid weight (g) and X the weight (kg) of the calf. As this calculation has only been validated in calves, we only compared the thyroid weights of calves using this formula.

The thyroid gland was removed within 12 hours of the animal's death. If the thyroid gland was not removed directly after the animal's death, the body was placed on tables, lying on their sides, at +4˚C in the necropsy room. To remove the thyroid gland, an incision was made through the skin with a scalpel (blade n˚20), cranial to the cricoid cartilage, and along the trachea for 10–20 cm. The sternohyoid and sternothyroid muscles were pulled away from the trachea, so that the trachea could be viewed. Once the trachea was visible, the isthmus of the thyroid was located, which is situated ventral to the cricoid cartilage, at the level of the second tracheal ring. The isthmus was grasped with a mouse-tooth pliers and the gland was dissected from the trachea to the right or left so that the lobes could be exteriorized. Once the thyroid gland had been removed, excess connective and fatty tissues that were not part of the gland itself were trimmed away. The thyroid gland was sampled at the slaughterhouse when operator removed the tongue-larynx-trachea-lung assembly: the thyroid gland was intact and was dissected directly on the slaughter line. All dissections were made by the same author.

After extraction of the thyroid gland, the left and right lobes were identified. *Post-mortem* measurements of the width, length, thickness of the lobes and thickness of the isthmus were taken using a 30 cm slat (accuracy 0.1 cm). To measure thickness, the gland was cut in half lengthways and the maximum thickness is measured with the same slat. This step was performed when the gland has no lesions visible on ultrasound. If one or more lesions were visible, the thyroid gland was first cut to fit over these lesions, and then cut to obtain the maximum thickness. The thyroid gland was weighed on an electronic balance (Ohaus Ranger 3000®, accuracy 0.5 g). Thyroid volume was measured using a container graduated up to 500 mL (accuracy 5 mL). The container was filled to 300 mL with mains water and the volume of the gland was measured by water displacement (Archimedes' principle).

Then, as soon as the measurements had been taken, the glands were macroscopically examined, and the areas where the lesions were visible, or corresponding to the location of the lesions on US, were macroscopically observed. When lesions were observed on US, the samples were focused on these lesions so that they appeared on the histological slide. Some thyroids pieces (1-2cm sides) were immersed in a jar containing formaldehyde (4% solution) to carry out a histological examination and detect any abnormal structures. A single histopathological section was taken. The samples were paraffin-embedded and then cut with a microtome (section between 4 and 5 μm). The stain used was Haematoxylin-Eosin. Histological slides were observed under the microscope at x40, x100, x200 and x400 magnification.

The statistical analysis was conducted using R software version 4.2.1 To assess the normality of the data distributions, a Shapiro test was performed on the various datasets. Once the normality was determined, statistical comparisons were made between US and *post-mortem* measurements using specific indices. The gHedges index (32) and Welch's t-test were employed to provide a comprehensive analysis by considering both the magnitude of the difference between groups (gHedges index) and the significance of that difference (Welch's t-test). To evaluate the agreement between the two measurement methods, a Bland-Altman graph was used. This graph plots the difference between the measurements made by the two methods on the x-axis, while the y-axis represents the average of the two methods' measurements. The information derived from this graph helped in assessing the level of agreement between the measurement methods. Lastly, correlations were calculated to examine the relationship between the volume calculation and the volume measured *post-mortem*, between the thyroid weight and the calculated thyroid weight in calves and between volume and weight of the

thyroid and age and weight of the calves. This correlation analysis provided insights into the strength of the relationships between these variables.

## Results

### 1. Population measurements

Twenty-two cows and 23 calves (16 males; 6 females) have finally been included in the protocol. Six Belgian Blue cows came from the slaughterhouse; 16 other cows came from the University Veterinary Clinic. All the calves included in the study came from the University Veterinary Clinic. The breeds of the cows were 15/22 Belgian Blue, 5/22 Holstein, 1/22 Limousin, 1/22 Blonde d'Aquitaine. They were 4.5±1 years old and weighed 656±148 kg (mean±SD). For calves, the breeds were 16/23 Belgian Blue cows, 5/23 Holstein, 1/23 Blonde d'Aquitaine and 1/23 Brown Swiss. Their age and weight were respectively 29±28 days and 56±17 kg. The causes of euthanasia for cows from the University Veterinary Clinic were peritonitis (7/16), pneumonia (3/16), obstructive digestive disease (2/16), traumatic reticulopericarditis (2/16), endocarditis (1/16), paratuberculosis (1/16) and for calves various digestive (11/23), locomotor (7/23) respiratory (4/23) and nervous (1/23) conditions. No pathology seemed directly or indirectly linked to a thyroid problem. The animals did not receive any treatment that could have influenced their thyroid status in the month prior to their death, such as injection of NaI for a wooden tongue, or injection of Vitamin E/selenium, or oral intake of T4 in calves or other unusual iodine/selenium supplementation, goitrogenic substances in the diet, etc.

### 2. Comparisons between US and *post-mortem* measurements, volume and weight estimation of the thyroid gland

The measurements made by US and *post-mortem* examinations are available in **Table 1**. No significant difference was found when comparing right and left lobes (Welch's t-test, p>0.1). Several comparisons were made between US and *post-mortem* measurements. Figs 1 and 2 illustrate the comparison between the measurements listed in Table 1. In calves, only the lengths were comparable between US and *post-mortem* measurements (p>0.1), but the effects of the size (gHedges) were small for these measurements, which means that the small sample size for these measurements could influence interpretation. Welch's t-test revealed that, in cows, none of the US measurements corresponded to the *post-mortem* measurements. The size effect was important for all measurements [30].

Regarding calves' measurements, the averages of thyroid measured weight (*post-mortem*) or calculated with the formula (37; $Y = 0.348 *X^{0.944}$) (Table 1) are significantly different (t-test; p<0.01). The correlation between these two parameters is 0.47 (p = 0.024). Correlations were also made between age, calf weight, thyroid weight and thyroid volume. Thyroid volume was correlated with calf weight (corr: 0.52; p value: 0.01), but not with calf age (corr: 0.13; p value: 0.55). Thyroid gland weight was neither correlated with weight (corr: 0.62; p value 0.19) nor with age (corr: 0.05; p value: 0.83).

When measured volumes (flotation) were compared with calculated volumes (US), there was no correlation. For cows, the coefficient of correlation was -0.021, and not significative (p > 0.05). For calves, they were almost correlated (p = 0.678) and significant (p < 0.05).

The average percentage of difference between the two methods is 57% for cows and 18% for calves. The variability between the two measurement methods is shown in Figs 3 and 4. It is observed that Fig 3, showing the differences in calves, is less scattered than Fig 4. In cows, Fig 4 also shows a systematic bias between the two methods even with a log2 transformation, whereas the transformation applied on calves' data is sufficient to mitigate this error (Fig 3).

**Table 1. Mean ± standard deviation (SD) of the measurements (cm) made at ultrasound (US) and *post-mortem* examinations in calves and cows.**

| | Calves (n = 23) | | Cows (n = 22) | |
|---|---|---|---|---|
| **US estimates** | LL | RL | LL | RL |
| **Height (cm)** | 1.13 ± 0.14 | 1.26 ± 0.37 | 2.53 ± 0.91 | 2.80 ± 0.97 |
| **Width (cm)** | 1.74 ± 0.39 | 1.62 ± 0.47 | 2.93 ± 0.65 | 2.88 ± 0.93 |
| **Length (cm)** | 3.17 ± 0.75 | 3.11 ± 0.78 | 8.32 ± 2.27 | 8.46 ± 2.46 |
| **Isthmus thickess (cm)** | 0.38 ± 0.20 | | 0.68 ± 0.19 | |
| **Volume calculation (VOL π/4; cm³)** | 10.30 ± 7.20 | | 108.10 ± 65.90 | |
| **Volume calculation (VOL π/6; cm³)** | 6.90 ± 4.80 | | 72.09 ± 43.90 | |
| **Expected thyroid weight calculation (g)** | 15.51 ± 4.32 | | - | |
| ***Post-mortem* measurements** | | | | |
| **Height (cm)** | 0.82 ± 0.32 | 0.76 ± 0.27 | 1.35 ± 0.37 | 1.35 ± 0.40 |
| **Width (cm)** | 2.62 ± 0.56 | 2.54 ± 0.46 | 4.62 ± 0.75 | 4.63 ± 0.72 |
| **Length (cm)** | 3.47 ± 0.41 | 3.17 ± 0.50 | 6.39 ± 1.51 | 5.98 ± 1.09 |
| **Isthmus thickess (cm)** | 0.28 ± 0.12 | | 0.47 ± 0.24 | |
| **Volume measurement (mL)** | 11 ± 10 | | 50.9 ± 19.3 | |
| **Thyroid weight (g)** | 12.62 ± 4.24 | | 51.90 ± 17.69 | |

LL = left lobe; RL = right lobe; VOL = volume formula.

This implies that the US method in calves is consistent with *post-mortem* measurements (with an average error of 18%).

## 3. Description of lesions observed on histological examination and their relation to US examination

Cystic lesions were found on US in 9 out of 22 cows. At *post-mortem* examinations, 9/9 thyroid glands had cysts, but histological lesions were found in only 7/9. Fig 5 shows the appearance of

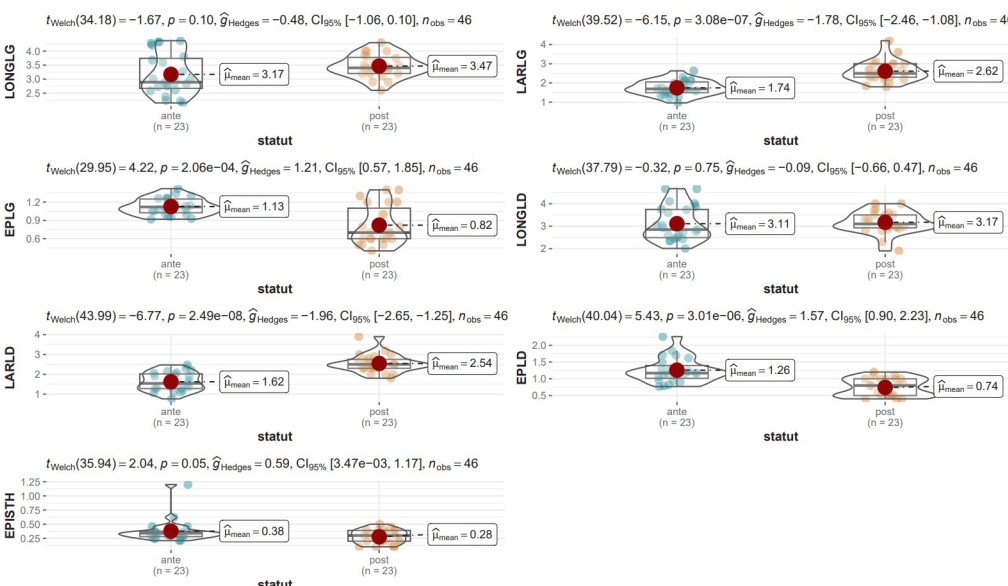

**Fig 1. Comparisons between ultrasound estimates and *post-mortem* measurements in calves.** For each measurement, gHedges, Welch t-test with p value, the confidence interval, the number of measurements, the distribution graph and the mean are available.

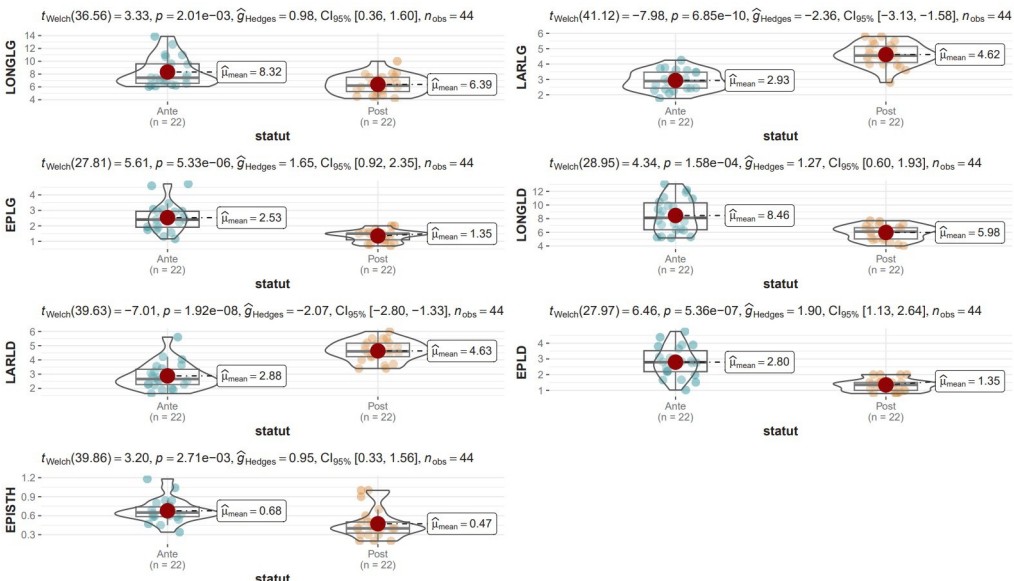

**Fig 2. Comparisons between ultrasound estimates and *post-mortem* measurements in cows.** For each measurement, gHedges, Welch t-test with p value, the confidence interval, the number of measurements, the distribution graph and the mean are available.

cysts on ultrasound and *post-mortem* in a cow. A histological view of the lesions seen in Fig 5 is also available in Fig 7. Some US and histological lesions observed in cows are described in Figs 6 and 7. The lesions were thyroid cyst (3/7 cows) (Figs 6 and 7) and follicular hypoplasia (7/7 cows) (Fig 7). The Fig 6 shows the appearance of a thyroid cyst 2 cm x 0.4 cm (diameter)

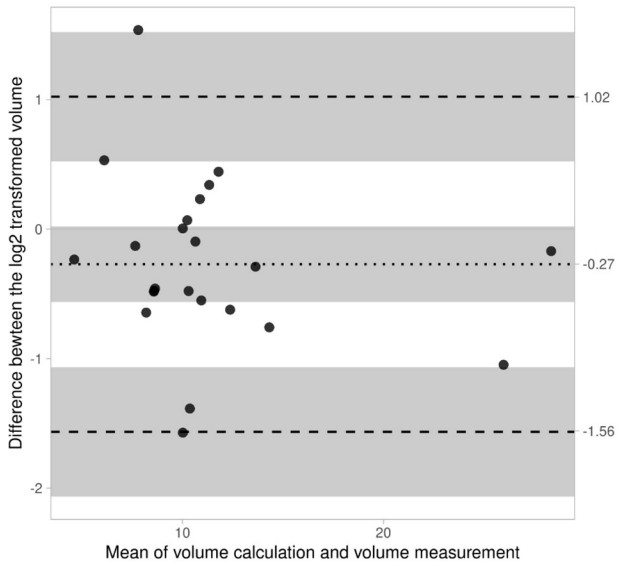

**Fig 3. Bland–Altman plot between ultrasound estimates and *post-mortem* measurements in calves.** Dashed lines represent the 95% confidence interval. Graph shows the data as dots. The dotted line indicates the average difference between log2 transformed data of -0.27 and the dashed lines indicate the Limits of Agreement: -1.56 & 1.02. The Shapiro-Wilk test of the difference indicates a p-value of 0.4692 which does support a normal distribution. The grey bands reflect the 95% confidence interval. The number of datapoints is 23.

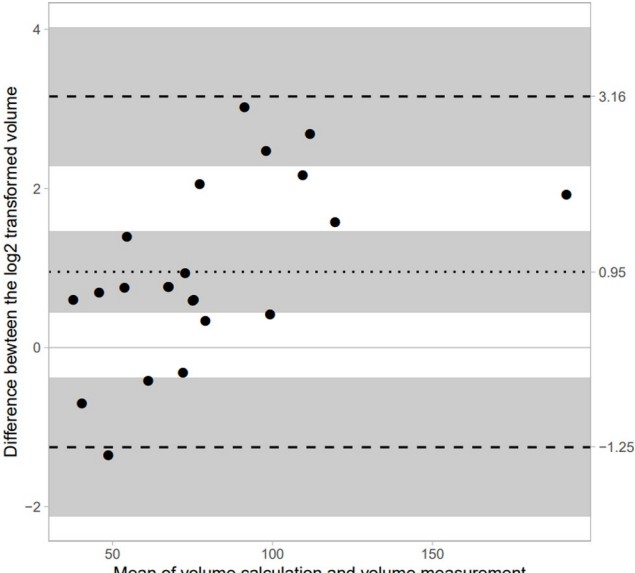

**Fig 4. Bland–Altman plot between ultrasound estimates and *post-mortem* measurements in cows.** Dashed lines represent the 95% confidence interval. Graph shows the data as dots. The dotted line indicates the average difference between log2 transformed data of 0.95 and the dashed lines indicate the Limits of Agreement: -1.25 & 3.16. The Shapiro-Wilk test of the difference indicates a p-value of 0.7031 which does support a normal distribution. The grey bands reflect the 95% confidence interval (CI). The number of datapoints is 22. The slope of the difference has 95% CI [0.01, 0.03] that does not include zero, suggesting a proportional bias.

on US. The other histological slides showed healthy tissue. Follicular hypoplasia is observed when the follicles are large and bordered partially by inactive epithelium. An inactive epithelium is flattened, the nucleus is centered in the cell and there is an absence of vacuolar vesicles at the apex of the cell, in the colloid. The presence of follicular hypoplasia is estimated as a percentage observed on the slide. A cyst is a round structure filled with liquid (colloid) that can be seen on ultrasound. When the cyst is cut on the histological slide, it is bordered only by an inactive, flat epithelium in places, and cubic epithelium in others.

No US lesions were observed in calves. Although no calf showed any lesions on US, 7 histological examinations were performed randomly on them. Two out of 7 histological examinations in calves revealed no lesions. The lesions seen on histology in the other 5 calves are further described. The lesions observed were follicular hyperplasia in one calf (Fig 8) and follicular hypoplasia (4/5 calves). Hyperplasia has been detected following the observation of an impression of increase in follicular cells, giving the appearance of the epithelial outline of follicles in "several layers of cells" and even deforming these follicles, which lose their ellipsoidal shape. Follicular hypoplasia is seen in calves according to the same criteria explained above for cows.

## Discussion

Thyroid US is a non-invasive, and relatively cheap examination of the thyroid gland, that can be performed without imposing an important stress in cattle [1, 5]. The examination can be routinely performed after a relatively easy training period using standard imaging material. It may bring not only morphological but also functional information regarding the endocrine status in these animals. Although used frequently in small animals (and humans) [6, 7, 10, 12], it is not yet considered as a routine procedure in cattle.

Indeed, the body size and sheer strength of these animals may render this procedure more difficult than in small animals and one may wonder about the accuracy of the measurements

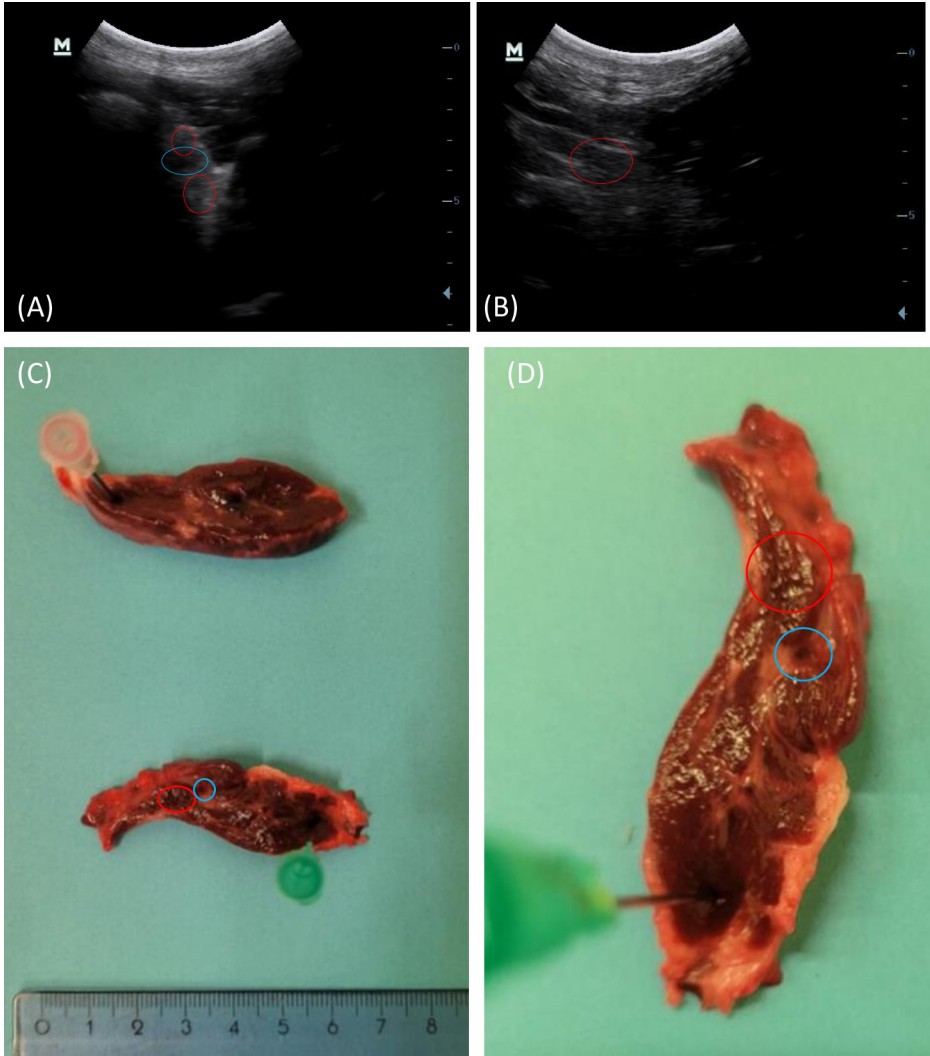

**Fig 5.** Ultrasound (A,B) and *post-mortem* pictures of the thyroid gland (C,D). The red and blue circles show the cysts visible on ultrasound, *post-mortem* examination and histology. The gland in *post-mortem* examination is cut in transversal section, where the cyst was supposed to be on ultrasound examination. The cyst circled in blue measures between 1 and 2 mm in diameter and is represented histologically in Fig 7 (A,B,C).

and the observations in bigger animals who have also a thicker fat layer that could impede the correct observations [5].

In a study published previously [5], we tried to assess the inter and intra-observer variability of thyroid gland measurements between three operators (one veterinarian radiologist and two veterinarians trained in cattle thyroid US). This first study showed that with some training, consistent measurements can be made, even in cows in which the procedure presents more technical difficulties. A next step for us was to compare in the present study, the in-vivo US measurement with *post-mortem* data of the same animals. These *in vivo* and *post-mortem* comparison was also used to assess the accuracy of the mathematical formulas that are used in the literature to calculate thyroid gland volume and weight, based on measurement made in three dimensions with US [7, 9, 29].

Although with practice, operators can achieve consistency in thyroid US measurements, significant discrepancies were noted between US and *post-mortem* measurements. Several

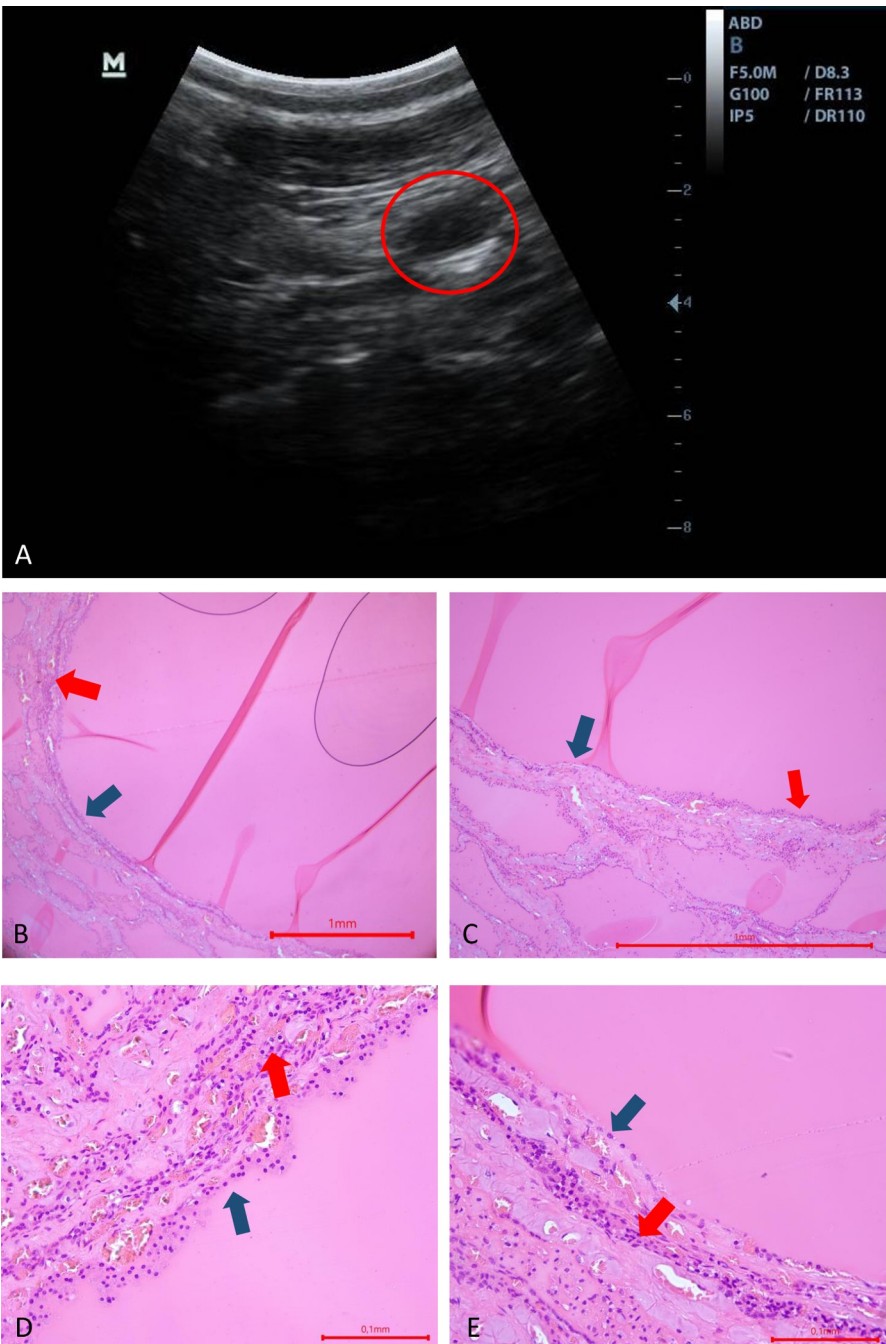

**Fig 6.** Ultrasound (A) and histological pictures, haematoxylin-Eosin stain. (B, C, D, E) of a follicular cyst 1 cm x 0.4 cm. (A) Sagittal view of the left lobe. The cyst appears as a structure bordered by a hyperechoic border and anechoic contents. (B) Magnification x 40, the cyst appears to be bordered by a simple epithelium (blue arrow) filled with pink-stained colloid. The follicle is surrounded by collagen fibers from the connective tissue (red arrow). (C) Magnification x 100, view of the cyst and the adjoining thyroid parenchyma. The thyroid follicles appear to be crushed by the follicle, and the slide shows around 90% follicular hypoplasia. The cyst is composed of inactive epithelium, flattened in places (blue arrow), and active epithelium in others (red arrow). (D) Magnification x400, this view shows a more cubic, active epithelium (blue arrow). The multi-layered appearance is attributed to an alteration in the sample and is rather artefactual (E) Magnification x400, this view shows the flattened epithelium (blue arrow). The nucleus is centered and no endocytosis vesicle is visible in the cytoplasm. The vascularisation around the follicle is abundant. In (D) and (E), collagen fibers can be seen organised concentrically around the cyst (red arrows).

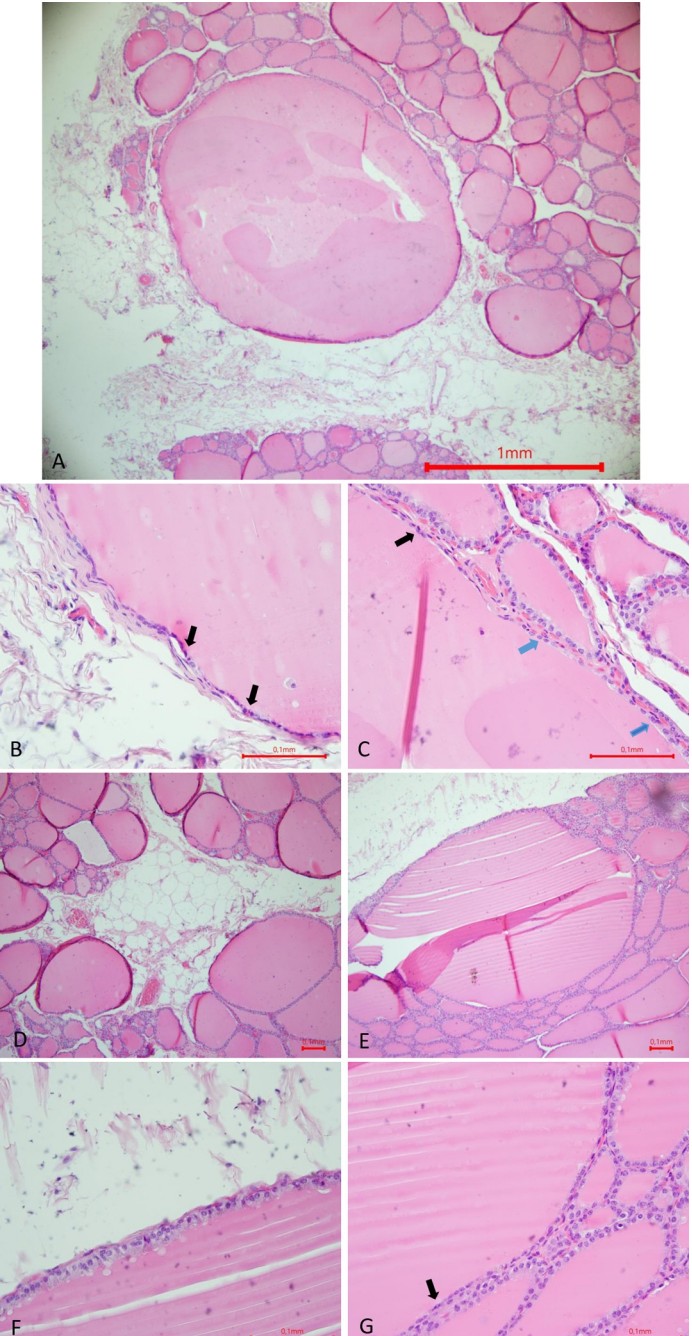

**Fig 7.** Histological pictures, haematoxylin-Eosin stain (A, B, C, D, E, F, G) of a follicular cyst (A, B, C), follicular hypoplasia (E, F, G) and fat infiltration (A, D) of cow's thyroids. (A) Magnification x 40, cyst in relation to the follicular tissue (D,E) Magnification x 100; (B,C,F,G) Magnification x 400. (A,B,C) The cyst is a large fluid-filled follicle whose epithelium is not homogeneously active: in places, the epithelium is flattened (B, C, black arrows), the central nucleus with condensed chromatin and the inactive cell containing little cytoplasm with fine, slightly eosinophilic granulations; in other places, the epithelium is cubic (C, blue arrows) with granular, slightly eosinophilic cytoplasm and a clear, decondensed nucleus. Compared with (E), in (A) the cyst is larger and detectable by US. (A,D) Appearance of a thyroid gland in a Belgian Blue cow at the end of fattening: infiltration of interstitial tissue by fat. Glandular tissue is more widely spaced than in the thyroid of a thin animal. There is no associated fatty degeneration. (E) Follicular hypoplasia, large follicles filled with colloid. Thyroid parenchyma is 60% hypoplastic on this slide. Once again, this follicle is made up of two populations of cells: ; rectangular active cells with a decondensed nucleus, an abundant granular, slightly eosinophilic cytoplasm with endocytosis vesicles; flattened inactive cells with a condensed nucleus, little cytoplasm with fine, slightly eosinophilic granulations (G).

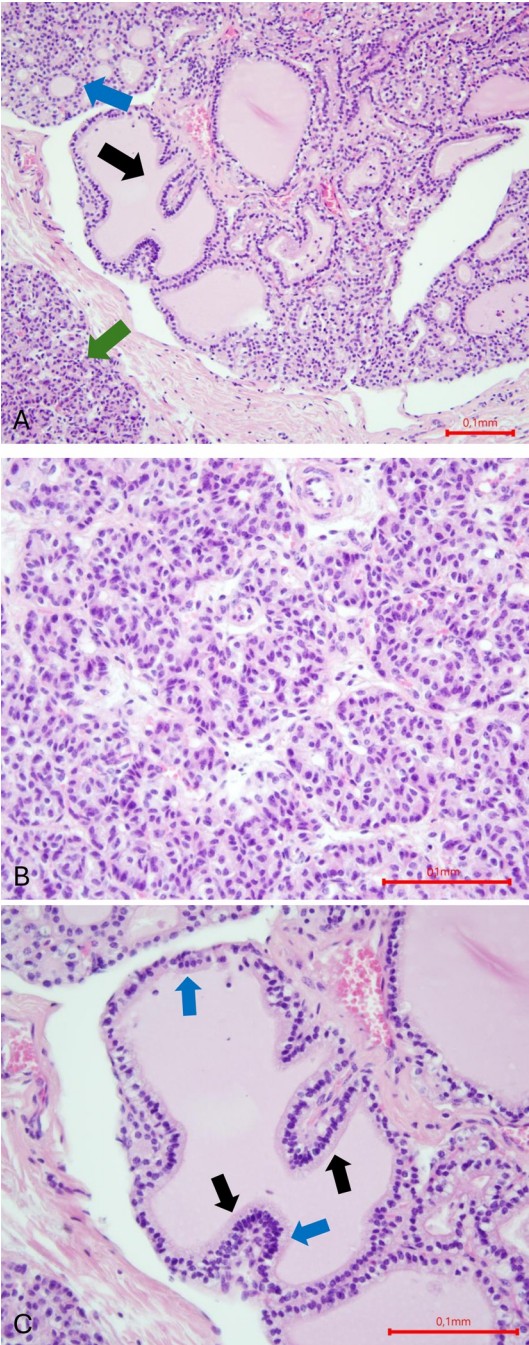

**Fig 8.** Histology of a hyperplastic thyroid gland in a calf, haematoxylin-Eosin stain (A, B, C). Follicular hyperplasia is characterized by an increase in the number of follicular cells. (A) Magnification x 200. The general appearance of the thyroid gland is irregular, and abnormally cellular. It is composed of follicles containing colloid with an irregular, flattened or elongated shape, with digitations (black arrow). Some follicles have a regular shape with colloid and an active epithelium (blue arrow). In other areas, the follicles are small, without colloid (green arrow). (B,C) Magnification x 400. (B) In the area containing the empty round follicles, the cells are cubic, the cell nuclei are decondensed and positioned at the base of the cell, and the cytoplasm is granular at the apical pole. (C) The follicle illustrated here has an irregular shape with digitations (black arrows). The epithelial cells are cubic, the nuclei are round, decondensed with clearly visible chromatin, sometimes with 1 or 2 nucleoli visible. The cells appear very tightly packed together, sometimes giving the impression of a multi-layered epithelium (blue arrows).

hypotheses may explain these differences The calf's thyroid is smaller, and the thyroid enters the entire field of the probe in all cases, so the probe does not have to be moved to measure the end of the thyroid's length. This is not the case in cows, where the translation of the probe may lead to a loss of accuracy while measuring dimensions. The very characteristic triangular shape of the thyroid gland in sagittal view makes it very easy to take measurements in calves, because the image is easy to recognize. This is not the case for the transversal view, where the maximum thickness of the isthmus must be measured before taking the other measurements, which could explain the difference in measurements between these two parameters in relation to length [5]. Moreover, the thyroid is a soft structure that is quite mobile and depending on the pressure exerted on the organ and its movement, it is possible to influence the shape and create measurement errors [5, 31]. On the other hand, *post-mortem* measurements are done in a gland that is outside of its normal anatomical enclosure. This may lead to modifications of its normal shape, changing the dimensions measured with a slate.

The calculated *in vivo* volumes did not correlate well with the measured volumes of the thyroid gland *ex vivo (*It should be noted that these calculations calculation gave better results in calves (coefficient of correlation: 0.678; $p<0.05$) than in cows). This is not surprising because human medicine provides similar conclusions [32, 33]. The same observation is made in dogs [10], and goats [29]. In another study, the volume measured (by flotation) was strongly related (correlation 0.98; $p<0.01$) to the volume calculated (US) in Metzner et al. (2015) in calves. On the other hand, in our study, there is an error percentage of 18% for calves and 57% for cows, which suggests that in the case of significant hypertrophy of the gland, it should be possible to differentiate it from a healthy gland, as this is the case in other species [10, 12, 34]. However, more work is needed to confirm this.

Volume immersion measurement can also be the source of some errors. Indeed, in some cows, the thyroid gland is embedded in fat, which is sometimes difficult to trim *post-mortem*. There may also be some measurement errors due to fluid displacement associated with this fatty deposit. There is also some interstitial fat in the thyroid gland that could influence this measurement, and the thyroid gland must always be completely immersed before measuring the volume. In a study of an obese child, it was found that the calculated volume of the thyroid gland increased with weight gain, and this increase in volume was reversible with weight loss. This increase/decrease in volume was not accompanied by functional thyroid disorders (blood tests with no abnormalities) [35]. Finally, it has previously been described that it is more difficult to visualize the thyroid gland in fat cows [5].

The weight of the calves' thyroids was predicted based on the formulas available in the literature [36] but did not match strongly with the measured weight (coefficient of correlation: 0.47; $p = 0.024$). One possible explanation is that the calves enrolled in the study were not confirmed euthyroid as thyroid laboratory tests (b-TSH, T4, T3, iodine) were not performed. Another explanation is that the formulas published in the literature estimates the normal weight of a thyroid gland in a healthy calf between 0–6 days but cannot be used to extrapolate the weight of the thyroid gland in sick or older calves. This study therefore did not invalidate this equation as a predictor of thyroid weight in calves. A thyroid gland weighing more than 30 g is considered abnormal [26] but in this present study, no gland exceeded this weight (maximal weight = 27.95 g).

In our study, a correlation between calf weight and volume (corr: 0.52; p value: 0.01) was found, but not between calf age and thyroid gland weight or volume, nor between calf weight and thyroid gland weight. The average thyroid measurements reported in this study were obtained in a selected population of calves and cows. Also, some animals were sick and scheduled to be euthanized. We believe that the data presented here should therefore not be used as reference values for normal population, util a higher number of subjects are studied.

Our findings showed the link between US-observable lesions (e.g. cysts) to a corresponding histological lesion (Fig 5). Some lesions discovered *post-mortem*, such as follicular hypoplasia or hyperplasia, were not detectable on US. More work is needed to see if other parameters such as echogenicity, elastography, or Doppler evaluation allow a better detection these lesions on US. It has been widely documented that hypothyroidism is characterized by hypoechogenicity of the thyroid gland [10, 12]. A comparison of thyroid echogenicity coupled with an analysis of blood hormone measurements may be useful.

Little is known in human medicine about thyroid cysts and their pathophysiology. There are cysts and pseudocysts [6]. A cyst can be seen visually, macroscopically on a gland at post-mortem examination or using an ultrasound scanner [6, 13]. A cyst is a round, transparent structure that contains liquid and is rougher than the rest of the thyroid tissue [13]. A cyst is a rounded, homogeneous fluid-filled structure (anaechogenic) surrounded by a hyperechoic capsule. Cyst is the most frequently lesion observed accidentally in children, with a prevalence of 60% [6, 37]. In human medicine, cysts are benign and require no special intervention. Nevertheless, a relationship has been described between insulin resistance syndrome and the presence of thyroid cysts and thyroid volume, with a prevalence of 45% of thyroid cysts in people with severe insulin resistance [37, 38]. In human, small cysts (< 3 mm diameter) may spontaneously disappear [39]. Polycystic thyroids may be associated with hypothyroidism, sometimes in association with excess iodine [40, 41]. Pseudocysts, on the other hand, are degenerated hyperplastic nodules containing injured cells, hemorrhage and necrosis accumulated in the parenchyma. They may contain microcrystals [6]. In our study, some cows also showed polycystic thyroids, but no pseudocysts were found. Our team has recently presented preliminary results indicating that cows with cysts have altered negative energy balance parameters compared with healthy cows without cysts ([42]. The body condition score (significantly lower in thyroid cyst population; p value <0.01), non-esterified fatty acids (NEFA; significantly higher in thyroid cyst population; p value <0.05) and selenium (trend to be lower in thyroid cyst population; p value = 0.056) were among the parameters modified. When comparing the percentage of thyroid cysts in the 5 farms, this percentage was significantly negatively correlated with the blood glucose/NEFA ratio of all cows examined in each farm (pearson correlation; r = -0.92; p value < 0.05) [42].

Histologically, the healthy thyroid gland is made up of follicles of different sizes, with the follicles on the periphery of the gland often smaller than those in the center [43]. The follicles are filled with colloid, surrounded by a lay er of active epithelium, where rectangular cells can be seen, with a nucleus at the base where mitotic figures are visible and an apex filled with a resorption vesicle. The follicles are surrounded by connective tissue concentrating the C cells [43, 44]. A thyroid cyst is surrounded by a layer of flattened, inactive or cubic follicular epithelial cells. It is filled with colloid [13]. In our study, the cysts observed in cows had the same histological structure (Figs 6 and 7). Although the literature describes the epithelium of cysts or hypoplastic follicles as inactive, on our slides, most of these epithelia have two cell populations, one active and one inactive. However, there are no rectangular cells in the cysts or hypoplastic follicles observed in this study. One study in two adult dairy cows describes a hyperplastic goiter associated with atrophy, fibrosis and cysts and attributes these lesions to chronic iodine excess and/or a history of iodine deficiency [14]. In our study, cysts with follicular hypoplasia were observed without hyperplasia, as already described in a study on buffalo [13]. Follicular hypoplasia or atrophy of the thyroid gland in dogs is caused by a decrease in serum b-TSH levels, either due to exogenous administration of thyroid hormones, or a defect in b-TSH secretion [45]. In ruminants, the other name is colloid goiter [13]. Histologically, the follicles are distended, the cellular epithelium is flattened and there is an absence of vacuolar resorption in the colloid at the epithelial margin [45]. Thyroid hyperplasia is the opposite: it is caused by

high levels of b-TSH or disturbances in the feedback mechanism of thyrotropin-releasing-hormone or b-TSH, and physiologically, indicates a sign of adaptation [26, 45, 46]. Hyperplastic thyroids are made up of areas containing normal follicles, areas containing macrofollicles, and areas with microfollicles, empty follicles that give the gland a very cellular appearance [46]. There is an impression of layers of supernumerary follicular cells, an increase of follicular cells, depletion of colloid in the follicles and papillary proliferations that deform the follicles [13]. In our study, the lesions observed on histological sections were consistent with those described in the literature (Fig 8).

In our study, US lesions could not be linked to histological examination in 2/9 cows. Indeed, it is difficult to locate the lesion *post-mortem* based on US examination, and it is also difficult to cut the right spot with a microtome. Moreover, cutting the thyroid gland *post-mortem* without first soaking it in formaldehyde can damage the cells that surround lesions such as cysts, and empty the cyst vacuole, making recognition of the structure more difficult. *In vivo*, additional examinations can be performed such as fine needle aspiration or needle aspiration biopsy [47]. Fine needle aspiration is the test of choice for an initial exploration of nodules, it is easy to perform, does not require anesthesia, and allows several thyroid tumors to be diagnosed [47]. For some cases (cyst >1cm), when fine needle aspiration in inconclusive, the use of ultrasound-guided needle aspiration biopsy should be studied, even if this technique presents false negatives in the case of multiple nodules [7, 48]. However, this technique requires a local anesthetic, a skin incision and a cyst of sufficient size to avoid missing it [47]. In the case of cysts it could be used to exclude other nodular pathologies.

It is interesting to know that in humans, the thyroid gland could increase its volume without functional problems in obese children and that hypoechoic areas have also been observed in children [35]. In addition, there are similarities in terms of duration of gestation between women and cows, as well as in the pathologies encountered, such as polycystic ovary syndrome [38, 49], which also involves thyroid function in women and in cattle. All these similarities lead us to believe that cattle could be a model for humans in terms of understanding thyroid metabolism, particularly the cyst pathology and pregnancy hormonal metabolism.

In calves, since volume measurement are correlated, the US could become a way to measure the thyroid gland as a follow-up to detect hyperplastic goiter at an moderate to severe stages, evaluate the benefits of supportive treatment with thyroxine and/or detect the restoration of the gland to normal values in calves suffering from acute respiratory distress [26, 27]. Further studies should be made to see if the same kind of relations can be made in cows using specific US data tables, despite the *in -vivo post-mortem* differences. Once again, observation of echogenicity or irrigation of the gland could be interesting parameters to investigate in the diagnosis of hypothyroidism and early stage hyperplastic goiter. The use of a Doppler makes it possible to visualize the irrigation and to differentiate a "hot" nodule from a "cold" nodule, which modifies the management in human medicine [6]. In addition to this tool, elastography is widely used in human medicine, and makes it possible to measure tissue hardness, with malignant nodules having greater hardness than benign ones [6]. More studies need to be applied on the use of these additional tools in rural medicine.

To the author's knowledge, the description of thyroid cyst lesions without the presence of goiter or hyperplasia in cattle has never been reported previously. It remains to be explained how these cysts and hypoplasia appear, and whether it is possible to link these lesions to a clinical metabolic state, for example the negative energy balance. Future work integrating complete blood tests including metabolic and thyroid parameters is necessary to understand the etiology of these lesions. The use of combined techniques, such as fine needle aspiration biopsy, should also be considered. Finally, it would be interesting to perform thyroid US on confirmed hypo-

or hyperthyroid animals to evaluate the performances of US in discriminating healthy and thyroid-affected animals.

## Conclusion

Although thyroid US cannot accurately measure the volume of the thyroid gland, this method has good agreement, especially in calves, with *post-mortem* examination of the gland, with an error percentage of 18% for calves and 57% for cows. This examination could be used to monitor the same animal, in a follow-up context, to check the changes in the volume of the thyroid gland. Further, it is possible to diagnose cystic lesions using this examination. Thyroid cysts in cattle have the same US characteristics as in humans, as well as the same histological structure. Additional work should focus on the etiology of thyroid cysts in cattle and the use of US in the diagnosis of thyroid pathologies and metabolic state (hypothyroidism, hyperthyroidism, negative energy balance), and the use of other related techniques such as fine needle aspiration biopsies.

## Supporting information

**S1 Data.**
(XLSX)

## Acknowledgments

The authors would like to thank the veterinarians, technicians and farmers who participated directly or indirectly in the successful completion of this study.

## Author Contributions

**Conceptualization:** Justine Eppe, Elise Raguet, Frédéric Rollin, Hugues Guyot.

**Data curation:** Justine Eppe, Elise Raguet, Sébastien Czaplicki.

**Formal analysis:** Justine Eppe, Hugues Guyot.

**Investigation:** Justine Eppe, Patrick Petrossians, Sébastien Czaplicki, Vinciane Toppets.

**Methodology:** Justine Eppe, Elise Raguet, Sébastien Czaplicki, Calixte Bayrou.

**Project administration:** Justine Eppe.

**Supervision:** Patrick Petrossians, Vinciane Toppets.

**Validation:** Patrick Petrossians, Hugues Guyot.

**Visualization:** Frédéric Rollin.

**Writing – original draft:** Justine Eppe, Sébastien Czaplicki, Calixte Bayrou.

**Writing – review & editing:** Patrick Petrossians, Calixte Bayrou, Frédéric Rollin, Vinciane Toppets, Hugues Guyot.

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
