## [Decision Letter · Decision Letter 0]

16 Feb 2024

PONE-D-23-43278From Ultrasound to Microscopy: actualities in thyroid investigation in cattlePLOS ONE

Dear Dr. Eppe,

Thank you for submitting your manuscript to PLOS ONE. After careful consideration, we feel that it has merit but does not fully meet PLOS ONE’s publication criteria as it currently stands. Therefore, we invite you to submit a revised version of the manuscript that addresses the points raised during the review process.

We look forward to receiving your revised manuscript.

Kind regards,

Benito Soto-Blanco, DVM, MSc, PhD

Academic Editor

PLOS ONE

Journal Requirements:

3. We note that your Data Availability Statement is currently as follows: [Toutes les données pertinentes sont contenues dans le manuscrit et ses fichiers d'information complémentaires.]

Reviewers' comments:

Reviewer's Responses to Questions

**Comments to the Author**

1. Is the manuscript technically sound, and do the data support the conclusions?

Reviewer #1: Yes

Reviewer #2: Yes

2. Has the statistical analysis been performed appropriately and rigorously? 

Reviewer #1: Yes

Reviewer #2: Yes

3. Have the authors made all data underlying the findings in their manuscript fully available?

Reviewer #1: No

Reviewer #2: Yes

4. Is the manuscript presented in an intelligible fashion and written in standard English?

Reviewer #1: Yes

Reviewer #2: Yes

5. Review Comments to the Author

Reviewer #1: Myself have reviewed this manuscript but some of the following questions have not been answered.

- The sexes of the calves are not the same. Or why there was no comparison between the sexes.

- The ages of the calves are not equal. Why is there no comparison according to their ages?

- Calves' body weights are very different. Why was there no comparison made according to body weights?

- The ages and body weights of the cows were not given. Therefore, there are doubts about the uniformity point. Can a comparison be made according to age and body weight?

- There is not enough information about the disease histories of cows and the treatment methods applied. This is also a factor that disrupts uniformity.

- Are there hematological and biochemical blood analyses in animals? Can parameters such as T3, T4 and TSH be given specifically?

Reviewer #2: The work is very interesting. I believe that the information generated will serve as a basis for future work to explore in depth and characterize thyroid diseases through non-invasive complementary methods. Currently there are few works that address this topic. It is very important that in the future the methodology be standardized in order to adequately compare ultrasound, macroscopic and microscopic pathology studies.

Line 37 - Hypothyroidism does not always generate enlargement or macroscopically detectable lesions. They refer

to the enlargement that occurs in hyperplastic goiter?

Line 101 to 103 - How was the thickness of the gland determined? Was a cut made in the gland? Or was a caliper used?

Some authors for example describe a cut in the middle part of each lobe to perform histological studies.

This same criterion could be useful to evaluate this parameter. Do the values of width, length and

thickness correspond to the measurement of a point? Because the gland is irregular in shape it should be

clarified at which point the measurement was made.

Line 110-111 - From which region was the section taken for histopathology? medial section, lateral section or medial

section (from the isthmus side)? Or was no fixed region taken and the presence of cysts was prioritized?

This should also be clarified in order to be able to make future comparisons since the size of the follicles and

the degree of activity of the gland may have differences associated with this.

Line 188 to 197 -

The paragraph could be reorganized. The findings of cows should be described first, accompanied by the

respective inserted image references. This should be followed by a description of the findings in calves

accompanied by their image reviews.

- Considering that the title of the paper indicates the focus on microscopic features, and that this aspect

comprises one of the main objectives of the paper, a more detailed description of the histological findings

or lesions present should be given. For example, detail what they define as hyperplasia: did they observe

stratification of follicular epithelium or papillary projections or both? A rough estimate in percentage could

be given. Likewise describe what they defined as follicular hypoplasia and what these findings consisted

of. It would also be good to define the characteristics of the cysts in terms of their lining epithelium,

whether it was flat or whether it consisted of cubic or columnar cells. Did they present foci of hyperplasia

in any region of the cyst?

- These observations seek to enrich the description since there are few works that describe in detail the

histopathological findings in bovine thyroid glands. I consider that a description should be included in the

body of the text in this section.

Line 198 to 200 - Detail in which direction the gland section was performed: sagittal, transverse, in the middle part of the

gland or there was no fixed region for all glands and it was performed looking for cysts. This could also

be detailed in materials and methods so as not to extend the text of the reference.

Line 203 to 204 - The image does not show the detail described. If it is present, it is an interesting description that should

be included in the text. What do they mean by vacuolar epithelium? Intracellular vacuoles? where are

they located? Do they displace the nucleus basally or apically?

Line 205 to 206 - What criteria do you use to define "hypoplastic parenchyma"? number of follicles? size? height of

epithelium?

Line 207 to 212 - What criteria do you use to define follicular cyst? What criteria do you use to define follicular hypoplasia?

It is important to set the criteria for defining hypoplasia to differentiate it from normal follicles of a gland

with little active tissue. It may simply be a matter of changing the term. Could a higher magnification

image be included to identify if the infiltration is in connective tissue septa or also appears between

follicles in a more diffuse manner? This is an interesting finding to describe and highlight. In human

medicine, fatty infiltration of the thyroid gland is considered pathologic in association with metabolic

diseases.

Line 288 to 290 - This statement is made using echogenicity features, they do not give details of histologic features. I

believe it would be valuable to detail their histopathologic observations in results. Updated works with

detailed descriptions of thyroid histology in cattle are scarce and this would be a great contribution.

Line 317 to 319 - The use of ultrasound-guided fine needle aspiration biopsy could not provide the information yielded by

the histologic study. Performing several sections to locate the cyst would help but it becomes expensive

and impractical to acquire as a routine method. An equivalent method should be sought to solve this

difficulty.

Line 327 to 330 - Considering that the thyroid gland does not always reflect alterations in the hypothyroid state, perhaps a

more adequate term for what they are trying to describe is hyperplastic goiter (this is a very common

pathology in ruminants, especially in calves and according to the degree of severity of the affection,

changes in the weight of the gland are generated. However, in the same way as mentioned above for the

weight of the gland, estimating the volume of the gland by ultrasound to detect abnormality, can be very

useful especially when there are goiter lesions, with moderate to severe degree (which generate a

detectable increase in the size of the gland). To detect goiter of mild grade or in early stages, due to the

error rate they have (error rate of 18%) it would not be possible to identify with certainty the

pathological state. I consider that changes in echogenicity or in the irrigation of the gland can be

interesting parameters to evaluate this aspect and circumvent this difficulty.

Line 343 - clarify that this statement is valid especially for calves.

Line 346 - The histopathological characteristics of thyroid cysts were not described in detail in the work.

Line 364 to 366 - Revise this citation.

6. PLOS authors have the option to publish the peer review history of their article (what does this mean?). If published, this will include your full peer review and any attached files.

Reviewer #1: No

Reviewer #2: **Yes: **Luis Adrián Colque Caro

---

## [Author Response · Author response to Decision Letter 0]

21 Mar 2024

5. Review Comments to the Author

Reviewer #1: Myself have reviewed this manuscript but some of the following questions have not been answered.

Thank you for your review

- The sexes of the calves are not the same. Or why there was no comparison between the sexes.

- The ages of the calves are not equal. Why is there no comparison according to their ages?

- Calves' body weights are very different. Why was there no comparison made according to body weights?

- The ages and body weights of the cows were not given. Therefore, there are doubts about the uniformity point. Can a comparison be made according to age and body weight?

We had already provided information on the weight and age of the cows in line 141-143 of the manuscript. They appear. As for the rest of your questions, we had indeed wanted to compare all this, but the number of animals in each category, breed and sex was very small, so the statistics had to be taken with caution and we preferred not to mention them in the paper. However, we have added correlations between calf age and weight, and thyroid volume and weight in lines 172-175.

- There is not enough information about the disease histories of cows and the treatment methods applied. This is also a factor that disrupts uniformity.

We have added information about the treatment on lines 147-150.

- Are there hematological and biochemical blood analyses in animals? Can parameters such as T3, T4 and TSH be given specifically?

Yes, we have results for plasmatic inorganic iodine, total T4 and other biochemical parameters such as urea, cholesterol, albumin, non-esterified fatty acids and fibrinogen. However, when we ran the statistics, nothing stood out because the population was too heterogeneous. We preferred not to mention this in the paper so as not to weigh it down. Finally, there is currently no validated b-TSH test for cattle in Europe, so it is impossible to provide this additional information.

Reviewer #2: The work is very interesting. I believe that the information generated will serve as a basis for future work to explore in depth and characterize thyroid diseases through non-invasive complementary methods. Currently there are few works that address this topic. It is very important that in the future the methodology be standardized in order to adequately compare ultrasound, macroscopic and microscopic pathology studies.

Thank you for your comment and for carefully reading our document. We're going to try to add more information to standardize the article, but it's difficult to be perfect in a field context.

Line 37 - Hypothyroidism does not always generate enlargement or macroscopically detectable lesions. They refer to the enlargement that occurs in hyperplastic goiter?

We completely agree with you. We didn't mean to imply that hypothyroidism systematically involves enlargement of the thyroid gland. We've changed the term in line 38 by hyperplastic goiter.

Line 101 to 103 - How was the thickness of the gland determined? Was a cut made in the gland? Or was a caliper used? Some authors for example describe a cut in the middle part of each lobe to perform histological studies. This same criterion could be useful to evaluate this parameter. Do the values of width, length and thickness correspond to the measurement of a point? Because the gland is irregular in shape it should be clarified at which point the measurement was made.

Yes, you're right, it wasn't clearly explained. Here's the explanation I've added in lines 104-107. To measure thickness, the gland is cut in half lengthways and the maximum thickness is measured with the same slat. This step is performed when the gland has no lesions visible on ultrasound. If one or more lesions are visible, the thyroid gland is first cut to fit over these lesions, and then cut to obtain the maximum thickness.

Line 110-111 - From which region was the section taken for histopathology? medial section, lateral section or medial section (from the isthmus side)? Or was no fixed region taken and the presence of cysts was prioritized? This should also be clarified in order to be able to make future comparisons since the size of the follicles and the degree of activity of the gland may have differences associated with this.

You're right, the sentence was badly worded. We've reorganised the paragraph into lines 112-113, so we hope it's clearer.

Line 188 to 197 - The paragraph could be reorganized. The findings of cows should be described first, accompanied by the respective inserted image references. This should be followed by a description of the findings in calves accompanied by their image reviews.

We have reorganised the entire paragraph into lines 200-264.

- Considering that the title of the paper indicates the focus on microscopic features, and that this aspect comprises one of the main objectives of the paper, a more detailed description of the histological findings or lesions present should be given. For example, detail what they define as hyperplasia: did they observe stratification of follicular epithelium or papillary projections or both? A rough estimate in percentage could be given. Likewise describe what they defined as follicular hypoplasia and what these findings consisted of. It would also be good to define the characteristics of the cysts in terms of their lining epithelium, whether it was flat or whether it consisted of cubic or columnar cells. Did they present foci of hyperplasia in any region of the cyst? These observations seek to enrich the description since there are few works that describe in detail the histopathological findings in bovine thyroid glands. I consider that a description should be included in the body of the text in this section.

You're absolutely right, we've added these details in the results section on the lines 245-264 and 375-379: “Hyperplasia has been detected following the observation of an impression of increase in follicular cells, giving the appearance of the epithelial outline of follicles in "several layers of cells" and even deforming these follicles, which lose their ellipsoidal shape”.

Line 198 to 200 - Detail in which direction the gland section was performed: sagittal, transverse, in the middle part of the gland or there was no fixed region for all glands and it was performed looking for cysts. This could also be detailed in materials and methods so as not to extend the text of the reference.

This has been added to the legend in lines 213-214.

Line 203 to 204 - The image does not show the detail described. If it is present, it is an interesting description that should be included in the text. What do they mean by vacuolar epithelium? Intracellular vacuoles? where are they located? Do they displace the nucleus basally or apically?

There was a misunderstanding in the description because this epithelium no longer has vacuoles when it borders a cyst, even though it is basically a vacuolar epithelium. We have removed the word "vacuolar" from the description and added, as mentioned above, a description of the histological appearance of a cyst in the main text. You can find the modifications in lines 217-229.

Line 205 to 206 - What criteria do you use to define "hypoplastic parenchyma"? number of follicles? size? height of epithelium?

As explained above, a description has been added in lines 206-209.

Line 207 to 212 - What criteria do you use to define follicular cyst? What criteria do you use to define follicular hypoplasia? It is important to set the criteria for defining hypoplasia to differentiate it from normal follicles of a gland with little active tissue. It may simply be a matter of changing the term. Could a higher magnification image be included to identify if the infiltration is in connective tissue septa or also appears between follicles in a more diffuse manner? This is an interesting finding to describe and highlight. In human medicine, fatty infiltration of the thyroid gland is considered pathologic in association with metabolic diseases.

For criteria to define follicular cyst and follicular hypoplasia, a description has been added as explain above in lines 206-209. As far as fatty infiltration is concerned, we observed all the slides of this thyroid gland carefully with a histologist (Vinciane Toppets, added in authorship), but the fat cells are really confined to the interstitial tissue. We did not observe fat lobules between the follicles or between the follicular cells. But thank you very much for your very interesting comment, because we pay attention to this in our next studies! We have added this information to the legend in lines 238-240.

Line 288 to 290 - This statement is made using echogenicity features, they do not give details of histologic features. I believe it would be valuable to detail their histopathologic observations in results. Updated works with detailed descriptions of thyroid histology in cattle are scarce and this would be a great contribution.

A description of the structure of the cysts on histology has been added in in a paragraph on histology lines 362-364.

Line 317 to 319 - The use of ultrasound-guided fine needle aspiration biopsy could not provide the information yielded by the histologic study. Performing several sections to locate the cyst would help but it becomes expensive and impractical to acquire as a routine method. An equivalent method should be sought to solve this difficulty.

You're right, we've added a sentence to qualify our comments in lines 385-391.

Line 327 to 330 - Considering that the thyroid gland does not always reflect alterations in the hypothyroid state, perhaps a more adequate term for what they are trying to describe is hyperplastic goiter (this is a very common pathology in ruminants, especially in calves and according to the degree of severity of the affection, changes in the weight of the gland are generated. However, in the same way as mentioned above for the weight of the gland, estimating the volume of the gland by ultrasound to detect abnormality, can be very useful especially when there are goiter lesions, with moderate to severe degree (which generate a detectable increase in the size of the gland). To detect goiter of mild grade or in early stages, due to the error rate they have (error rate of 18%) it would not be possible to identify with certainty the pathological state. I consider that changes in echogenicity or in the irrigation of the gland can be interesting parameters to evaluate this aspect and circumvent this difficulty.

We have respected your comments in lines 400 and 404-410.

Line 343 - clarify that this statement is valid especially for calves.

We've added it in lines 422-42.

Line 346 - The histopathological characteristics of thyroid cysts were not described in detail in the work.

This has been added as mentioned above in lines 200-264 and 358-379.

Line 364 to 366 - Revise this citation.

Thank you for spotting this error, it has been changed in lines 445-447.

---

## [Decision Letter · Decision Letter 1]

5 Apr 2024

From Ultrasound to Microscopy: actualities in thyroid investigation in cattle

PONE-D-23-43278R1

Dear Dr. Eppe,

We’re pleased to inform you that your manuscript has been judged scientifically suitable for publication and will be formally accepted for publication once it meets all outstanding technical requirements.

Kind regards,

Benito Soto-Blanco, DVM, MSc, PhD

Academic Editor

PLOS ONE

Reviewers' comments:

Reviewer's Responses to Questions

**Comments to the Author**

1. If the authors have adequately addressed your comments raised in a previous round of review and you feel that this manuscript is now acceptable for publication, you may indicate that here to bypass the “Comments to the Author” section, enter your conflict of interest statement in the “Confidential to Editor” section, and submit your "Accept" recommendation.

Reviewer #1: All comments have been addressed

Reviewer #2: All comments have been addressed

2. Is the manuscript technically sound, and do the data support the conclusions?

Reviewer #1: Yes

Reviewer #2: Yes

3. Has the statistical analysis been performed appropriately and rigorously? 

Reviewer #1: Yes

Reviewer #2: Yes

4. Have the authors made all data underlying the findings in their manuscript fully available?

Reviewer #1: Yes

Reviewer #2: Yes

5. Is the manuscript presented in an intelligible fashion and written in standard English?

Reviewer #1: Yes

Reviewer #2: Yes

6. Review Comments to the Author

Reviewer #1: (No Response)

Reviewer #2: (No Response)

7. PLOS authors have the option to publish the peer review history of their article (what does this mean?). If published, this will include your full peer review and any attached files.

Reviewer #1: No

Reviewer #2: **Yes: **Luis Adrian Colque Caro
